# Analysis of the Influence of the Geometrical Parameters of the Body Scanner on the Accuracy of Reconstruction of the Human Figure Using the Photogrammetry Technique

**DOI:** 10.3390/s22239181

**Published:** 2022-11-25

**Authors:** Maciej Trojnacki, Przemysław Dąbek, Piotr Jaroszek

**Affiliations:** 1EDUROCO sp. z o.o., ul. Łąkowa 3/5, 90-562 Łódź, Poland; 2ŁUKASIEWICZ Research Network—Industrial Research Institute for Automation and Measurements PIAP, Al. Jerozolimskie 202, 02-486 Warsaw, Poland

**Keywords:** human figure, 3D model, 3D scanner, human body scanner, HUBO scanner, photogrammetry, scanning accuracy, numerical analysis, experimental research

## Abstract

This article concerns the research of the HUBO full-body scanner, which includes the analysis and selection of the scanner’s geometrical parameters in order to obtain the highest possible accuracy of the reconstruction of a human figure. In the scanner version analyzed in this paper, smartphone cameras are used as sensors. In order to process the collected photos into a 3D model, the photogrammetry technique is applied. As part of the work, dependencies between the geometrical parameters of the scanner are derived, which allows to significantly reduce the number of degrees of freedom in the selection of its geometrical parameters. Based on these dependencies, a numerical analysis is carried out, as a result of which the initial values of the geometrical parameters are pre-selected and distribution of scanner cameras is visualized. As part of the experimental research, the influence of selected scanner parameters on the scanning accuracy is analyzed. For the experimental research, a specially prepared dummy was used instead of the participation of a real human, which allowed to ensure the constancy of the scanned object. The accuracy of the object reconstruction was assessed in relation to the reference 3D model obtained with a scanner of superior measurement uncertainty. On the basis of the conducted research, a method for the selection of the scanner’s geometrical parameters was finally verified, leading to the arrangement of cameras around a human, which guarantees high accuracy of the reconstruction. Additionally, to quantify the results, the quality rates were used, taking into account not only the obtained measurement uncertainty of the scanner, but also the processing time and the resulting efficiency.

## 1. Introduction

Research work on the development of human body scanners has been carried out for many years, which has resulted in the creation of many solutions. An overview of this type of solution can be found, among others, in the publication [1], where these solutions were divided into three categories, including passive stereo, structured light and time-of-flight. In turn, on the website [2] there is a list of over 500 scanning systems, including 44 full-body scanner solutions.

Due to the widespread availability and low cost of smartphones, they are becoming more and more popular as a tool for the reconstruction of objects using the photogrammetry technique. The paper [3] presents the results of research on the reconstruction of a dummy head using this approach. In turn, the article [4] is devoted to the method of measuring the parameters of the human figure, also with the use of smartphones and the photogrammetry technique. Smartphones as a tool for collecting data about scanned objects were also used in the works of [5,6].

Scanning systems based on the photogrammetry technique are of interest to many industries. In the field of human scanning, it is possible to mention, among others, virtual fitting rooms [7], fitness and sport [8], education [9], as well as health and medicine [10,11,12,13,14]. Typically, as a result, the scanning of stationary models has been obtained, although attempts have also been made to create interactive models, which may be used, for example, in the education of anatomy [15].

In the standard approach, the scanning of a human figure is carried out in a stationary pose. In the case of automated scanners, it takes from a fraction of a second to a few minutes, and in the case of manual scanners, from a few to several dozen minutes. In addition, if one wants to get the dimensions of the figure based on a 3D model, it is most often recommended to scan in underwear. The problem with human scanning is discussed in [16], in which it was noted that a human is significantly more complex than still life, both in terms of photogrammetric capture, and in digital reproduction. In turn, work [17] describes a system based on the use of 100 cameras for human reconstruction based on the photogrammetry technique.

More recently, attempts have also been made to scan in motion, which is the subject of, among others, the article [18], which discusses the state of the art in this field and presents preliminary research results. The problem of scanning a human in motion is also analyzed in the work [19], in which the so-called 4D scanning system (Move 4D) is based on the photogrammetry technique. In this case, a set of synchronized measurement modules was used, each of which consists of a pair of infrareds (IR) cameras, an IR projector, a color (RGB) camera and a processing unit.

An important question of a practical nature is, how accurate are the measurements with scanning systems compared to manual measurements? As indicated in the paper [20], when using automated measurement based on 3D scanners, the measurement uncertainty is usually superior than in the manual method compliant with ISO 20685. In the case of some of the dimensions of the figure discussed in this publication, the measurement uncertainty was initially worse, but after taking into account the constant offset, better results were achieved in each case, compared to the manual method. A similar subject is addressed in the article [21], which presents the results of the research which showed that 3D scanners provide reproducible, accurate and reliable data that correlate well with the different techniques (including manual) used for the evaluation of body measurements. In turn, work [22] presents the results of the investigation in the field of measurement accuracy of 3D body scanning mobile applications, also comparing them with the traditional manual measuring method. On the basis of the conducted research, it was found that the investigated applications provide acceptable measurement accuracy of the human body in the analyzed application.

Solutions based on vision systems nowadays use the photogrammetry technique, the state of the art and development directions which are described in [23]. The conference publication is also devoted to the review of the state of the art in the field of photogrammetry [24]. There are also attempts to combine photogrammetry with other scanning techniques to take advantage of each of them. For example, in [25], the photogrammetry technique was combined with photometric stereo, achieving high-quality reconstruction. In publications [26,27], data from laser scanning and photogrammetry were integrated. In turn, the article [28] proposes the method of the forensic 3D documentation of bodies based on combining CT scanning with photogrammetry.

The photogrammetry technique is based on a three-dimensional reconstruction of an object on the basis of a set of photos. Therefore, the amount of obtainable information from the photos has a key influence on the results. High quality images having large resolutions, good sharpness and detailed patterns of contrasting features are preferable. In addition, the accuracy of the reconstruction can be increased, e.g., through a histogram adjustment of images, which is the subject of the work [29].

The reconstruction process using photogrammetry technique can be performed in the following stages (see Figure 1):1.Generating a sparse point cloud, including:a.finding common features (tie points) in different images,b.a generation of camera poses based on common features between the images;2.Densifying the point cloud based on all pixels of all images;3.Generating a multi-polygon model (mesh of triangles) based on the point clouds from previous stages;4.The generation of textures for the multi-polygon model.

The first stage can start when a set of photos is given. It consists of determining the camera parameters, including their poses in space, as well as intrinsic parameters like focal length or distortion model. Note that the initial values of the camera intrinsic parameters can be obtained in the optional procedure of camera calibration (for example, from multiple photos of a checkerboard), which may improve reconstruction accuracy. A mandatory task is finding the features within each photo, e.g., scale-invariant feature transform (SIFT) keypoints, which are crucial from the point of view of automatic feature extraction and matching, which was emphasized in the work [30]. Then, the features are matched across the entire image set so that for each feature a subset of images where the feature is observed is formed. By calculating a fundamental matrix between the first image pair taken from the subset, it is possible to determine poses of cameras and then triangulate positions of points in space associated with the features. Having the first image pair and the first set of points in space, from now on the algorithm of finding subsequent camera poses is switched to a kind of Perspective-n-Point algorithm. For the newly found camera pose, the positions of points in space associated with the features are triangulated. Reprojections of the feature points in space on camera images are calculated. Errors between original and reprojected points on camera images are calculated in terms of the pixel distances. The adjustment of all determined camera parameters is performed to minimize the overall error. The procedure is iteratively carried out until no further camera poses can be added. This method is called “structure from motion” (SfM) [31] and its output is the sparse point cloud.

In the second stage, with known camera poses and parameters, a dense cloud of points is generated. This is done with the aid of “multi-view stereo” (MVS) algorithms, which enable the calculation of depth values for all pixels of an image. For each image a depth map is calculated, and depth maps for images from different camera poses are combined together on the condition of keeping the consistency between multiple cameras. This stage requires large computational effort, which is correlated with the resolution of the images (overall number of pixels), but the computations can be usually performed in parallel. For the points in the dense cloud, their colors can be computed if necessary.

In the third stage, the dense point cloud is transformed into a mesh of triangles (a 3D model that consists of vertices and faces). There are several algorithms for surface reconstruction that can be used and they generally fall into two categories: volumetric approaches, and surface-based approaches [32]. There are also methods that can reduce the size of the 3D model and at the same time preserve the model shape and improve mesh topology, especially for models with large number of redundant vertices.

In the last, fourth stage, color textures are generated for the model. In the process, called unwrapping, the 3D mesh of triangles is mapped on a 2D plane that contains color pixels (an image) so that it is known which pixels belong to which triangle. Because in this stage additional graphic files with textures are produced, this stage is sometimes neglected if the color is not of interest, from the point of view of the model application.

Available software can be used to process photos using the photogrammetry technique. The publication [33] compares three such programs, that is, Agisoft Metashape, Bentley ContextCapture and RealityCapture, when applied to the reconstruction of cultural heritage objects. The Meshroom program, based on the popular AliceVision framework [34], is also one of the applications that perform all the processing steps discussed earlier. In turn, the article [35] also includes comparison of the performance of free photogrammetry software for reconstruction, taking into account Agisoft Metashape, 3DF Zephyr and Regard 3D.

It can be noticed that the major disadvantage of the photogrammetry technique, in relation to the laser (time of flight) technique, is computationally intensive and time-consuming data processing. Therefore, one should strive to minimize the number of photos necessary to prepare a complete 3D model of the scanned object, and to optimally distribute the photos in space to obtain an even accuracy of this model. Regardless of this, easily scalable cloud computing services can be used. Such solutions should ultimately speed up processing to meet the maximum time requirement, which ideally should not be longer than the time taken by an automated 3D scanner to capture photos.

Lighting is a key aspect influencing the result of scanning using the photogrammetry technique. In order to obtain an accurate model, it is necessary to evenly and constantly illuminate the scanned object and eliminate the so-called hard shadows. Adequate lighting is also important in terms of exposure time to avoid blurry photos, which is especially important if the photos are taken while cameras or human are moving. It is also important that the photos are taken from different directions, but in such a way that the viewing directions of the cameras are as perpendicular to the scanned planes as possible, because in this case the highest accuracy of reconstruction of the object is obtained.

When analyzing the available literature in the field of photogrammetry, one can notice few publications, such as [36], that contain precise guidelines regarding the distribution of camera poses of scanning systems around the scanned object. However, the distribution of cameras is a key aspect that influences the complete, accurate and efficient determination of the 3D model of the scanned object. By the complete model is meant a model that has all its regions properly reconstructed, without missing parts, and preferably without holes and other artifacts. In turn, one can talk about the efficient model determination when a minimum number of photos is used to provide a complete model of the required accuracy. It can be noticed that the processing time increases significantly as the number of photos increases, so the number of photos should be minimized on condition the required accuracy is maintained.

Therefore, the aim of this study is to develop the method of selection of the geometrical parameters of the scanner, including those related to the distribution of cameras, to obtain a complete and efficient full-body model of a human of the highest possible accuracy, or in the most efficient way, with an acceptable accuracy of the model.

## 2. HUBO Scanner

The subject of the research is the HUBO scanner by the EDUROCO company, shown in Figure 2a, i.e., a portable and automated system for scanning the human figure. The scanner has a modular design, as shown in Figure 2b. It consists of three main modules detachable for transport, which are a base, a platform attached to it, on which a person should stand, and a rotating mast with sensors.

The detailed structure of the scanner is shown in Figure 3. The support module includes a body (2), supporting set (3) equipped with extendable legs, a rotating arm (4) and an attached AC adapter (14). The platform module (1), on which a person should stand, is attached to the body (2). Relative to the body (2) turns the rotating arm (4), to which the mast module, consisting of the arms (6) and (7), is attached. The rotating arm (4) is driven by a drive unit with a DC motor. Due to the use of an electric rotary joint (slip ring) in the body (2), the rotating arm (4) can turn in an unlimited way. The mast module can be folded for transport thanks to the yokes (5) and (8). These yokes also make it possible to adjust the inclination of the arms (6) and (7) to the needs, in particular to the height of the scanned object. The arms (6) and (7) are equipped with trolleys (9) moving along them and driven by stepper motors, the wires of which are guided by cable carriers (16). The sensors (13) are attached to the trolleys (9) by means of connectors (12), which enable the collection of measurement data. In the analyzed case, smartphone cameras are used as sensors (13). Due to the servo-drives used in the trolleys (9), the sensors (13) can also be tilted with respect to the arms (6) and (7) by a desired angles. The mast arms are also equipped with LED lamps (11) attached with connectors (10), which enable the proper lighting of the scanned object. All scanner devices, including its controller (17), are powered by an AC adapter (14) attached to the body (2). In the upper part of the lower arm (6), an optional presence sensor (15) can also be placed, allowing the detection of the scanned object on the platform (1).

The construction of the scanner enables automatic taking of a series of photos of the scanned object, in the given poses of the cameras around it. On this basis, the software determines a 3D model of the scanned object using the photogrammetry technique. In the current version, the HUBO scanner uses the AliceVision framework [34] for this purpose.

The scanner is characterized by such main functionalities as the automation of the scanning process, integrated light source for lightening the scanned object, the configurability of the working space, as well as detecting the presence of the scanned object on the platform.

## 3. Analysis of the Scanner’s Geometrical Parameters

The key aspect of this work is the selection of geometrical parameters of the scanner, including those connected with the distribution of the cameras on it, which will ensure the completeness of the model of the scanned object, as well as the highest possible accuracy of the reconstruction. It is assumed that the scanner is configured in a way suitable to scan people with the maximum height H=2.1 [m] and that the configuration of the scanner does not change when scanning people with a height lower than the maximum. In the case of much shorter people, such as children, it is possible to limit the number of camera poses, by omitting the cameras in the highest positions.

The basic geometrical parameters of the HUBO scanner and the scanned human are illustrated in Figure 4.

Because of the possibility of movement of the trolleys, the cameras can be placed arbitrarily along the scanner mast in k positions, where k is even, since it is assumed that the number of camera positions relative to each of the two scanner arms is the same in the standard case.

Moreover, it is assumed that during single turn of the mast, the positions of the cameras relative to the arms do not change, but they can change between consecutive turns. That is, in the intervals between the consecutive turns, the trolleys with cameras are moved and the tilt of the cameras is also set in order to obtain the desired poses of them during the upcoming turn.

The position of the trolley with the camera relative to the lower arm is marked $a_{i}$, for i=1,…,k/2, and for the upper arm $b_{j}$, for j=1,…,k/2. The position of the trolley is understood as the distance between the center of the joint of the trolley, to which the camera is attached, and the axis of the lowest yoke of a given arm, measured along this arm (see Figure 4).

The index i is also used in the context of all camera or trolley positions, i.e., considering both arms, and then i=1,…,k. One can notice that in such case j=i−k/2.

The lowest positions of the trolleys are $a_{min}$ and $b_{min}$, whilst the highest positions $a_{max}$ and $b_{max}$, respectively for the lower and upper arm.

The angles of deviation from the vertical are $\gamma_{1}$ and $\gamma_{2}$, respectively for the lower and upper arm, where a positive angle is measured outside the scanner and a negative angle is measured inside.

The pitch angles of the cameras $\theta_{i}$ are measured against the horizontal, according to the principle that a positive angle is measured upwards, and a negative one downwards. In turn, the camera pitch angles with respect to the lower and upper arm are denoted $\alpha_{i}$ and $\beta_{j},\;\mathrm{respectively}$. They are measured with respect to the axis perpendicular to these arms, according to the same principle as the angles $\theta_{i}$ when it comes to the signs of the angles.

It can be seen that the pitch angles of the cameras $\theta_{i}$ depend on the angles $\alpha_{i}$ and $\beta_{j}$ as well as the angles of deviation of the arms from the vertical $\gamma_{1}$ and $\gamma_{2}$, according to the following relationships:(1)$$\theta_{i}=\{\begin{array}{ccc} \alpha_{i}+\gamma_{1} & \mathrm{for} & i\leq k/2,\\ \beta_{i-k/2}+\gamma_{2} & \mathrm{for} & i>k/2. \end{array}$$

Taking into account the geometrical limitations of the scanner, the minimum and maximum heights at which the trolleys with cameras can be positioned above the platform are respectively:(2)$$z_{min}=h+a_{min}\cos\gamma_{1}+c_{1}\sin\gamma_{1}$$
(3)$$z_{max}=h+l_{1}\cos\gamma_{1}+f_{1}\sin\gamma_{1}+b_{max}\cos\gamma_{2},$$
where $l_{1}$—length of the lower mast arm, h—distance of the lower mast yoke axis from the platform, $c_{1}$—shift of the trolley’s joint center from the lower arm longitudinal axis, $f_{1}$—shift of the upper yoke axis from the lower arm longitudinal axis.

Assuming that the trolleys with cameras are evenly distributed vertically during scanning, the vertical distances between their adjacent positions are:(4)$$\Delta z=\left(z_{max}-z_{min}\right)/\left(k-1\right)$$

Hence the height of the trolley’s joint i with the camera above the platform results from the dependence:(5)$$z_{i}=z_{min}+\left(i-1\right)\Delta z$$
for i=1,…,k.

The heights *z_i_* can also be expressed as a function of the geometrical parameters of the lower and upper arms in the form:(6)$$z_{i}=\{\begin{array}{ccc} h+a_{i}\cos\gamma_{1}+c_{1}\sin\gamma_{1} & \mathrm{for} & i\leq k/2,\\ h+l_{1}\cos\gamma_{1}+f_{1}\sin\gamma_{1}+b_{i-k/2}\cos\gamma_{2} & \mathrm{for} & i>k/2. \end{array}$$

By transforming the above relationship, it is possible to determine the positions of both trolleys in relation to the lower and upper arm, respectively, from the relationship:(7)$$a_{i}=\left(z_{i}-h-c_{1}\sin\gamma_{1}\right)/\cos\gamma_{1}\;\mathrm{for}\;i\leq k/2$$
(8)$$b_{i-k/2}=\left(z_{i}-h-l_{1}\cos\gamma_{1}-f_{1}\sin\gamma_{1}\right)/\cos\gamma_{2}\;\mathrm{for}\;i>k/2$$

In turn, the distances $r_{i}$ of the trolleys from the mast rotation axis can be described as a function of geometrical parameters of the scanner using the formula:(9)$$r_{i}=\{\begin{array}{ccc} L+a_{i}\sin\gamma_{1}-c_{1}\cos\gamma_{1} & \mathrm{for} & i\leq k/2,\\ L+l_{1}\sin\gamma_{1}-f_{1}\cos\gamma_{1}+b_{i-k/2}\sin\gamma_{2} & \mathrm{for} & i>k/2, \end{array}$$
where L is the distance of the lower yoke axis from the mast rotation axis.

Knowing the distances $z_{i}$ and $r_{i}$, it is possible to determine the pitch angles of cameras resulting in different views on selected parts of the object depending on adopted viewpoints distribution strategy. For this purpose, it can be assumed that the object has height H divided into k−1 parts that have successive heights $h_{i}$ on the z axis.

The simplest strategy is to assume a uniform change in viewpoint height $h_{i}$, e.g., from 0 to H with step H/(k−1). Then, the desired viewpoint height can be calculated from the formula:(10)$$h_{i}=H\left(i-1\right)/\left(k-1\right)$$

Due to the specificity of the shape of the human figure, it may be beneficial, however, to distribute the viewpoint height unevenly. For example, the head and hands are characterized by greater geometrical complexity, while the legs are quite simple geometrically. In addition, the face is the most important when it comes to the perception of a 3D model by a human.

One may also decide that the lower and upper parts of the scanned object should not be very close to the edge of the photo to avoid the risk of unsuccessful reconstruction of these parts of the object, for example, due to lens distortion.

In the general case, successive viewpoint heights $h_{i}$ can be calculated as a fraction or a percentage $dh_{i}$ of the height of the scanned person *H*, i.e., based on the relationship:(11)$$h_{i}=dh_{i}\;H$$

In turn, the angles describing the viewing directions of individual cameras at viewpoints with given heights $h_{i}$, i.e., camera pitch angles $\theta_{i}$ can be calculated from the following equation:(12)$$\left(h_{i}-z_{i}\right)/r_{i}=\mathrm{tg}\theta_{i}$$
where, as indicated earlier, the angles $\theta_{i}$ are measured from the horizontal.

The camera pitch angles are therefore equal to:(13)$$\theta_{i}=\mathrm{arctg}\left(\left(h_{i}-z_{i}\right)/r_{i}\right)$$

In practice, the camera is offset with respect to the trolley to which it is attached, and this offset can be divided into the component perpendicular to the arm, indicated in Figure 4 as length e and also in the component along the arm, which is indicated as f. The camera displacement relative to the trolley for the known lengths e and f can also be described in the radial direction and along the z axis based on the relationship:(14)$$dr_{i}=-e\;\cos\theta_{i}+f\;\sin\theta_{i}$$
(15)$$dz_{i}=e\;\sin\theta_{i}+f\;\cos\theta_{i}$$

Taking the above into account, the equations describing the pose of the camera pointing at the object at the height $h_{i}$ have the following form:(16)$$r_{i}=\left(e+g_{i}\right)\cos\theta_{i}+f\sin\theta_{i}$$
(17)$$z_{i}=h_{i}+\left(e+g_{i}\right)\sin\theta_{i}-f\cos\theta_{i}$$
where $g_{i}$ is the distance of the camera from the z axis at the height $h_{i}$.

Based on the above equations, the following more precise solution is obtained with regard to the camera pitch angle and its distance from the z axis at the height $h_{i}$:(18)$$\theta_{i}=\mathrm{arctg}\left((h_{i}-z_{i}-f\frac{w_{2}}{w_{1}})/r_{i}\right)$$
(19)$$g_{i}=r_{i}/\cos\theta_{i}-f\mathrm{tg}\theta_{i}-e$$
where $w_{1}=r_{i}\sqrt{{\left(z_{i}-h_{i}\right)}^{2}+r_{i}^{2}-f^{2}}-f\left(z_{i}-h_{i}\right)$ and $w_{2}={\left(z_{i}-h_{i}\right)}^{2}+r_{i}^{2}$.

The pitch angles of the cameras $\theta_{i}$ in their individual positions in relation to the scanner arms (defined by distances $a_{i}$ and $b_{j}$) depend on the angles of their inclination in relation to the lower and upper scanner arm, i.e., $\alpha_{i}$ and $\beta_{j}$, respectively.

Knowing the angles $\theta_{i}$ and the angles of deviation of the arms from the vertical $\gamma_{1}$ and $\gamma_{2}$, it is possible to determine the camera pitch angles in relation to the arms in their individual positions, based on the formulas:(20)$$\alpha_{i}=\theta_{i}-\gamma_{1}\;\mathrm{for}\;i\leq k/2$$
(21)$$\beta_{i-k/2}=\theta_{i}-\gamma_{2}\;\mathrm{for}\;i>k/2$$
where these angles are control variables in the HUBO scanner and are limited by the angular range of the servo-motors to ±π/4.

Knowing the poses of the cameras, it is possible to determine the point of intersection of the camera view axes with the z axis from the formula:(22)$$z_{vi}^{0}=z_{i}+dz_{i}+\left(r_{i}+dr_{i}\right)\;\tan\theta_{i}$$

Knowing additionally the vertical view angle of a camera $\Delta_{v}$, it is possible to calculate the intersection points with the axis z of straight lines defining the range of camera visibility, i.e., from the following dependencies:(23)$$z_{vi}^{min}=z_{i}^{\;}+dz_{i}-\left(r_{i}+dr_{i}\right)\;\tan\left(\theta_{i}-\Delta_{v}/2\right)$$
(24)$$z_{vi}^{max}=z_{i}^{\;}+dz_{i}+\left(r_{i}+dr_{i}\right)\;\tan\left(\theta_{i}+\Delta_{v}/2\right)$$

On the basis of the derived relations between the geometrical parameters of the scanner, a method of selecting values of these parameters was developed, which is illustrated in Figure 5.

The proposed method allows to significantly reduce the number of geometrical parameters of the scanner for which values need to be selected, and to adjust these values to the height of the scanned person. The values of the remaining parameters, thanks to the relations derived above, can be calculated on the basis of selected values of primary parameters. Moreover, this method can be implemented in a series of steps described below. In each step of the method, selected parameters are refined.

The first step consists in the selection of values for the constant geometrical parameters of the scanner and the initial selection of values for the remaining parameters for further optimization. The choice of these values is influenced primarily by the maximum size of the scanned object and the viewing angles of the cameras. The minimum lengths of the mast arms should be assumed, which are sufficient for scanning of objects within the assumed size range.

The second step is related to the selection of the maximum distance between the mast and its axis of rotation, as well as the optimal angles of its arms deviation from the vertical. On the one hand, it is necessary to limit the working space required by the scanner, and on the other hand, to adjust the angles of the arms to the envelope of the figure of a person of nominal height.

The third step is to select the optimal camera viewpoint distribution strategy to ensure the highest possible accuracy (or strictly the smallest measurement uncertainty) of the reconstruction for a given number of camera poses.

The fourth step is related to the analysis of the influence of the number of camera poses around the object, and along the mast (vertically) on the measurement uncertainty. The minimum number of camera poses should be selected which guarantees the necessary accuracy of the reconstruction. Changing the number of camera poses analyzed so far may require re-optimization of the camera viewpoint distribution strategy, i.e., a return to the third step.

The fifth step is related to the final determination of parameters related to the field of view of cameras, and the distances of individual cameras from the assumed shape of the human figure.

The final step covers the visualization of the geometrical configuration of the scanner for the final parameters and 3D models obtained as a result of data processing.

The derived method was used at the stage of laboratory research.

## 4. Laboratory Studies

The aim of the laboratory studies was to select such geometrical parameters of the scanner and the poses of its cameras using the proposed method, and to obtain the highest possible accuracy of the reconstruction of the scanned object in relation to the 3D reference model. In order to achieve this goal, an analysis of the influence of selected scanner parameters on the scanning accuracy was carried out.

The scope of the research covered:A.The selection of geometrical parameters of the scanner, including:a.the distance of the mast from the axis of rotation L,b.the angles of the mast deviation from the vertical for the lower and upper arms, i.e., $\gamma_{1}$ and $\gamma_{2}$, respectively;
B.The selection of the optimal arrangement of cameras on the scanner arms, including:a.the number of cameras, taking into account number of positions around the scanned object n and number of positions along the mast arms k,b.the positions of the cameras on the mast arms $a_{i}$ and $b_{j}$,c.the pitch angles of the cameras relative to the mast arms $\alpha_{i}$ and $\beta_{j}$.


The selection of parameters from groups A and B determines the location of the camera poses in the space around the scanned object. Therefore, the most important thing is not the geometrical parameters of the scanner, but how the poses of the cameras are arranged. However, in terms of the parameters of groups A and B, it can be noticed that by using the previously derived relations between them, i.e., by introducing additional criteria, it is possible to limit the number of degrees of freedom of these parameters. In particular, parameters B.b and B.c can be selected on the basis of parameters A.a, A.b and B.a.2, depending on the adopted view distribution strategy, for example, in such a way that for the assumed height of the scanned object H an even vertical distribution of the scanner cameras is ensured, as well as a similar overlap of individual photos and similar minimum camera distances from human figure.

The minimum distance of a camera is understood here as the shortest distance from the parts of the human body seen by the camera. However, the minimum distance at which each camera must be located from the human body is not analyzed in the present work. Its value was assumed, given eight to 10 cameras in the vertical direction, to guarantee the overlap of the photos such that all photos in the dataset are taken into account in the “structure from motion” algorithm by the AliceVision framework, and its default setup. In order to obtain a sufficient overlap of the photos, the cameras must be located sufficiently far from the human figure anyway. It should be noted that the greater the distance of the camera from the human figure, the larger the scanner’s working area and the lower the number of available pixels for the reconstruction of a given part of the object. However, this problem can be solved by increasing the resolution of the cameras.

### 4.1. Numerical Analyzes

Before starting the experimental research in laboratory conditions, a numerical analysis in a simulation environment developed in Python was carried out based on the geometric dependencies described in the previous section, which allowed to limit the number of different settings (variants) of scanner parameters being the subject of experimental research. The simulation environment also made it possible to visualize the distribution of cameras and the fields of view of the cameras.

The values of the constant geometric parameters of the HUBO scanner, presented in Table 1, were adopted for the analysis. In addition, the height of the scanned object was assumed H=1.775 [m], which corresponds to the height of the test dummy.

The following geometrical parameters were considered as variables at the stage of numerical analysis: k, n, L, $\gamma_{1}$, $\gamma_{2}$. Given the values of those parameters, the remaining geometrical parameters of the scanner and related to the poses of the cameras were determined, including: $r_{i},\;z_{i}$, $\theta_{i},a_{i}$, $b_{j}$, $\alpha_{i}$, $\beta_{j}$.

To make the analysis more realistic, a human figure was defined in the form of a scalable set of points in the xz and yz planes of the scanner coordinate system, as illustrated in Figure 6.

Knowing the approximate shape of the human figure, it is possible to calculate the minimum distance between the cameras and the scanned human figure $d_{i}$ in the area of the camera field of view. It can be noticed that, in practice, the human figure points in the yz plane are usually closest to the cameras (see Figure 6).

As a result of an example analysis for the initial set of scanner parameters: k=8, n=32, L=0.809 [m], $\gamma_{1}=5\;\left[{}^{\circ}\right]$, $\gamma_{2}=-15\;\left[{}^{\circ}\right]$ the geometrical parameters characterizing the distribution of the scanner cameras around the scanned object were obtained, as presented in Table 2. In this example the values of the coefficients describing cameras’ settings $dh_{i}$ were: ${\left[0.1,\;0.3,\;0.5,\;0.6,\;0.7,\;0.8,\;0.9,\;0.9\right]}^{T}$.

In Figure 7 are illustrated the yz plane of the scanner coordinate system, the results of this example numerical analysis, that is, the human figure, the positions of the trolleys joints, as well as the axes of view of the cameras, the ranges of their views and the minimum distances of the cameras from the human figure $d_{i}$.

### 4.2. Experimental Research

As part of preliminary experimental studies, the optimal angular velocity of the scanner’s rotating arm was selected. On the one hand, this velocity should be as high as possible so that the scanning time is as short as possible, but on the other hand, it must enable the taking of photos during the movement with the appropriate quality in terms of data processing in order to obtain a 3D model. In particular, it should be taken into account that too high a motion velocity of the camera may cause the blurring of photos, and too high an acceleration can lead to unintended dynamic effects resulting from the action of the inertia forces. Finally, the angular acceleration of the scanner’s rotating arm was selected in such a way that after its acceleration to the desired angular velocity, all photos were taken while moving at a constant velocity. The adopted angular velocity allows for a single scan to be performed in approximately T=170 [s] for k=8 positions of the cameras on the scanner mast.

The scanned object at the stage of experimental research in laboratory conditions was a female dummy specifically adapted for this purpose. The height of this dummy was H=1.775 [m] and its width was $W=2r_{0}=0.545\;\left[m\right]$. Its arms and legs were immobilized so that they do not move during subsequent tests. In addition, the entire surface of the dummy was covered with a light diffusion coating and a pattern that can be described well with SIFT (scale-invariant feature transform) keypoints.

For the dummy prepared in this way, a reference model was made using the highly accurate Hexagon Stereo Scan (AICON) scanner, shown in Figure 8, i.e., with a declared accuracy of ±40 [μm]. This reference model was further compared with the 3D models obtained from the tested HUBO scanner.

In order to compare the 3D model from the HUBO scanner with the reference model, over 100 measurement points were selected on the surface of the reference model. These points were then compared with the corresponding closest points of the tested 3D models, obtained as a result of scanning with the HUBO scanner.

The comparisons were made on the basis of 3D models in the form of a dense cloud of points. After matching both 3D models, i.e., the reference model and the tested model, the accuracy of the object reconstruction was determined for all measurement points.

The accuracy of 3D models obtained from the HUBO scanner was assessed on the basis of the following methodology:
1.The tested 3D model was transformed into the reference 3D model in such a way that the sum of squared errors in the selected characteristic points of these models was the smallest. This transformation included 3-element vectors taking into account translations, rotations and scale change.2.Coarse errors (so-called artifacts) whose absolute value was greater than five times the standard deviation determined of the mean for all characteristic points were removed.3.The accuracy of the model was determined in accordance with [37] as the standard uncertainty of type A, calculated statistically from the set of observation values as the standard deviation of the mean, i.e., for all characteristic points with the omission of the so-called artifacts.

The measurement uncertainty of the HUBO scanner was calculated for all *N* measurement points as the standard deviation of the mean σ from the formulas:(25)$$\sigma =\sqrt{\sum_{i=1}^{N}{\left(e_{i}-\mu \right)}^{2}/N},\;\mu =\sum_{i=1}^{N}e_{i}/N,$$
where μ is the mean error.

The tests were carried out for selected scanner configurations, determining 3D models under M=4 scanning trials under the same conditions and processing with the same reconstruction software settings. From among the recorded results for individual trials within a given variant, the three most similar ones were finally selected.

Experimental studies were carried out for the best configurations of L, $\gamma_{1}$ and $\gamma_{2}$ parameters selected at the stage of numerical analyzes. When it comes to the distance of the lower yoke axis from the mast rotation axis, a compromise value was adopted, i.e., L=0.809 [m]. Increasing the distance of the mast from the axis of rotation is disadvantageous due to the increased space required to perform scan. On the other hand, decreasing the distance leads to a reduction in the overlap of photos and may require the use of more cameras to reconstruct the entire object or cameras with a wider field of view.

The angles of deviation of the mast arms from the vertical were also finally selected for the experimental tests. These angles, equal to $\gamma_{1}=5\;\left[{}^{\circ}\right]$ and $\gamma_{2}=-15\;\left[{}^{\circ}\right]$, were initially selected at the stage of numerical analyzes. Such selected angles ensure similar distances of all cameras from the human figure. They also do not cause an excessive increase of the scanning space, which occurs for larger values of angle $\gamma_{1}$ of the lower arm of the mast.

As part of the experimental research, various options related to the placement of cameras were analyzed, including:(a)The vertical distribution of camera viewpoints (on the z axis);(b)The pitch angles of cameras;(c)The number of camera positions around the scanned object n;(d)The number of camera positions along the mast (vertically) k.

The analyzed variants of the research are further marked using the above designations of options (i.e., a–d), together with the consecutive number of the sub-options (e.g., a1).

In the research, the Xiaomi Redmi 8 smartphones were used to take photos, equipped with cameras with a maximum resolution of 3016 px × 4032 px. For these smartphones, the horizontal and vertical view angles were experimentally determined, equal to about $\Delta_{h}=33\;\left[{}^{\circ}\right]$ and $\Delta_{v}=61\;\left[{}^{\circ}\right]$, respectively. The cameras were not subject to any field calibration procedure and the camera intrinsic parameters were estimated alone by the “structure from motion” algorithm of the AliceVision framework with its default setup.

During the research, the maximum resolution of photos was used, as well as reduced twice to 1508 px × 2016 px. It has been observed that the use of maximum resolution does not improve the accuracy of the 3D model significantly. Moreover, the processing time in this case is at least twice as long. Therefore, the results presented in this paper refer to a reduced resolution.

#### 4.2.1. Research in Terms of the Vertical Distribution of Camera Viewpoints

When it comes to the vertical arrangement of the camera viewpoints, i.e., the coordinates of the points of intersection of the camera axis with the mast rotation axis (option a), basically, two cases were analyzed. The first variant a1 consisted in evenly distributing these points along the height of the object (see Figure 9(a1)).

The second variant, a2, was related to the different vertical distribution of the viewpoints of the cameras, so that they were adjusted to the specificity of the figure (see Figure 9(a2)), i.e., there were smaller distances between these points in the most important parts of the figure (such as the head, breasts and hands).

In both cases, the distribution of these points was defined by the previously described coefficients $dh_{i}$.

The values of the coefficients determining the camera settings for the variant a1 were $dh={\left[0.08,\;0.2,\;0.32,\;0.44,\;0.56,\;0.68,\;0.8,\;0.92\right]}^{T}$, while for the variant a2 they were equal to $dh={\left[0.08,\;0.28,\;0.48,\;0.6,\;0.68,\;0.76,\;0.84,\;0.92\right]}^{T}$.

The results of the experimental tests, i.e., the obtained 3D models for particular variants illustrated in Figure 9, are shown in Figure 10.

Comparing the test results for variants a1 and a2, and taking into account all trials, it can be noticed that for the variant a1 significantly worse results were obtained. For this variant, large unevenness of the surface of the model is visible, especially in the area of the breast and the outer parts of the arms (see Figure 11). Moreover, in this instance, the chin was not properly reconstructed. In the case of the model for the a2 variant, slight unevenness of the surface on the rear parts of the arms can be noticed (see Figure 11). At the same time, despite the fact that fewer cameras were directed to the legs in variant a2, no reduction in the smoothness of the model surface in these areas was observed.

#### 4.2.2. Research in Terms of Camera Tilt Angles

In terms of camera tilt angles (option b), basically, two strategies were analyzed. The first strategy was to select the camera tilt angles in such a way that subsequent cameras were directed at successive viewpoints. According to this strategy, it was permissible that two neighboring cameras were aimed at the same point.

The second strategy was that cameras that are not neighboring could be aimed at the same point. Points near the most important parts of the body were selected as common points for several cameras. The use of this strategy was motivated by the fact that when looking at a given part of the body only from one side, especially if it is geometrically complex, it is possible not to notice some of its essential features. For example, if a given camera looks at the face slightly from above, the lower part of the chin is invisible to it, which may lead to incorrect reconstruction of the human head.

In both strategies, the pitch angles of the cameras resulted directly from the adopted sequence of viewpoints, defined by the coefficients $dh_{i}$. Moreover, at this stage, efforts were made to select the best configurations of the coefficients $dh_{i}$ among those analyzed at the earlier stage, i.e., within option a.

Figure 9(b1) illustrates the camera settings and visibility for the b1 variant, while Figure 10(b1) shows the resulting 3D model. For this variant, the following values of camera setting coefficients were adopted: $dh={\left[0.1,\;0.3,\;0.5,\;0.6,\;0.7,\;0.8,\;0.9,\;0.9\right]}^{T}$.

In turn, Figure 9(b2) shows the settings and visibility of the cameras for the variant b2, and Figure 10(b2) shows the obtained 3D model. In this case, the following coefficients related to the camera settings were used: $dh={\left[0.1,\;0.3,\;0.5,\;0.5,\;0.7,\;0.9,\;0.7,\;0.9\right]}^{T}$.

The comparison of the research results for variants b1 and b2 indicates that in both cases quite good models were obtained. In the case of the b2 variant, an inaccurate reconstruction in one trial can be observed in the rear part of the right arm (see Figure 12) and a less smooth surface of the posterior part of the head in relation to variant b1. More visible differences between these variants can be noticed by performing a quantitative analysis, which will be discussed later in the article.

#### 4.2.3. Research in Terms of the Number of Camera Poses

When it comes to research in terms of the number of camera poses, the values of the parameter n related to the number of camera poses around the scanned object, were considered in the range from 16 to 32 (option c).

As for the parameter k related to the number of camera poses along the mast (option d), the numbers ranging from 8 to 10 were analyzed. The smaller number of cameras vertically would be insufficient due to the vertical view angle of the cameras and the necessary overlap of adjacent photos. At this stage of the research, efforts were made to take into account the best variants in terms of the arrangement of cameras from among those analyzed in the two previous stages of research.

The number of photos taken by the scanner during one scanning session resulted from the assumed number of camera poses around the scanned object n and along the mast k. Therefore, it was equal to n·k and ranged from 128 to 320 photos.

For the research on the arrangement of cameras around the scanned object, i.e., for option c, variants b1 and b2 were taken into account as reference. These variants correspond to the scanner configuration for n=32 and within this range of tests they are denoted as c1 and c2, respectively. Based on the recorded data for these variants, new variants were created for the number of cameras around the object n=16 and they were called c3 and c4, respectively. In this case, every second photo from variants c1 and c2 was taken into account.

Figure 13 shows the resulting models for variants c3 and c4, respectively. Comparing these results to the results for variants c1 = b1 and c2 = b2, at first glance, there is no significant reduction in the quality of the models. However, looking more closely at the models for the c3 and c4 variants, a slight reduction in the reconstruction accuracy in some areas of the object can be noticed as compared to the corresponding c1 and c2 variants.

For example, in the case of the model for variant c1 the cuts are deeper under the armpits (it means the reconstruction is better), while for variant c3, due to the smaller number of cameras around the dummy, these areas are less accurately reconstructed (compare variants c1 and c3 in Figure 14). In addition, for the c3 variant, a less accurate reconstruction of the chin can be noticed.

Within the last research option d discussed in this article, the influence of the number of camera positions on the mast arms (vertically) defined by the parameter k was analyzed. The variant d1 = b2 for k=8 was taken into account as a reference. Under variant d2, a similar camera arrangement strategy was analyzed but for k=10, as illustrated in Figure 15. For this variant, the coefficients $dh={\left[0.1,\;0.3,\;0.5,\;0.6,\;0.7,\;0.8,\;0.9,\;0.7,\;0.8,\;0.9\right]}^{T}$ defining camera settings were adopted. As a result, the model shown in Figure 15 was obtained.

When visually comparing the models for both variants d1 and d2, it can be seen that, on the one hand, some areas are more precisely reconstructed for the d2 variant, but on the other hand, in other areas there are larger surface unevenness compared to the d1 variant. Thus, these results in terms of accuracy seem comparable, but it can be assessed more objectively within a quantitative analysis.

Of course, if the goal is to achieve an accurate reconstruction of individual body parts, and processing time is not critical, then configurations with more cameras should be selected. Figure 16 shows the result of the reconstruction of one of the more complex parts of the body, which is the hand, on the example of variants c1 (n=32, k=8), c3 (n=16, k=8) and d2 (n=32, k=10). Comparing the results for these variants, one can see the highest accuracy of the reconstruction for the d2 variant characterized by the highest number of cameras.

#### 4.2.4. Quantitative Analysis of the Results

For the above-described variants of this research, the correctness of the determination of poses of cameras around the scanned object was verified, which was the result of the “structure from motion” algorithm. As a consequence of the operation of this algorithm, usually a full set of camera poses was obtained, i.e., equal to the number of taken pictures.

Then, a quantitative analysis of the results was carried out, in accordance with the previously described methodology. The analyzed model was fitted to the reference model and the measurement uncertainty was calculated for the analyzed set of points and particular trials within variant in the form of the standard deviation of the mean σ.

Figure 17 shows an example of the results of a dummy scanning with a HUBO scanner using the photogrammetry technique, i.e., the distribution of cameras around the scanned object in a perspective view, resulting from the operation of the "structure from motion" algorithm, as well as the example result of comparison of the cloud points with the reference model (for variant b2).

During the experimental studies, the value of the root mean square error (*RMSE*) parameter provided by the AliceVision framework was also analyzed. The *RMSE* parameter in this case results from the operation of the “structure from motion” algorithm and is a measure of the camera poses error calculated in the image plane for all SIFT keypoints and for all observations (camera poses). The *RMSE* error relates to 1 pixel.

Taking into account the image resolution used within the standard scanner settings, the average distance between the cameras and the object, approximately 0.5 [m], as well as the measured viewing angles of the cameras, the size of 1 px in the photo was about 0.25 [mm].

Table 3 shows the results of the quantitative analysis in terms of *RMSE* parameter and measurement uncertainty obtained for all the above-described analyzed variants of the experimental tests and for the most representative trial. The measurement uncertainty is described by the standard deviation of the mean σ. The results for the remaining trials were similar to those presented in Table 3, thus these results are representative and repeatable.

Table 3 also presents the average processing time T of datasets for individual variants. The calculations were performed on a workstation with the following specification: Threadripper 1950X processor with 16 cores and 3.4 GHz, RAM 64GB/3600MHz, GeForce RTX2070/8GB graphics card, ASUS Prime X399-A motherboard. It was also possible to take advantage of cloud computing to speed up calculations. However, a solution with a workstation was decided due to the greater possibility of viewing the processing at its every stage.

In order to assess the efficiency, i.e., the accuracy of the reconstruction in relation to the processing time, the efficiency coefficient *EF* was introduced. The value of this coefficient was calculated for each variant as the product of the standard deviation of the mean σ and the processing time T. It is assumed that this coefficient is related to 1 mm and 1 min, that is, it is assumed to be dimensionless. The lower the *EF* value, the greater the processing efficiency for the scanner configuration analyzed in a given test variant.

#### 4.2.5. Conclusions from the Comparative Analysis of the Results

By analyzing the values of the obtained quality rates, it can be concluded that the best results in terms of reconstruction accuracy were obtained for the variant b1 = c1. Good results were also obtained for variants c3, a2 and d2. It is worth emphasizing that the c3 variant, derived from the b1 = c1 variant, allowed for a good reconstruction accuracy with half the number of photos compared to the reference model. For variant a2, the values of two measures are better than the values obtained for variant d2 and one is worse. Due to the larger amount of data (photos), in the case of the d2 variant, a longer processing time is required, therefore the a2 variant seems to be better, i.e., for a smaller number of cameras.

As for the number of camera poses around the object, for the parameter n=16, satisfactory results were obtained in terms of measurement uncertainty for all measurement points, which can be seen e.g., by comparing the results for variants c2 = b2 and c4. While examining the models, one can notice slightly worse results in terms of the accuracy of the reconstruction of the inner parts of the arms, which results from the worse visibility of these areas by the cameras. Thus, the overall model accuracy score was only slightly better for n=32, but was associated with twice as many photos and a significantly longer data processing time.

It should be noted that the parameters k and n can be arbitrarily increased, but for practical reasons, it is worth adopting the maximum values above which no further significant improvement in the accuracy of the model is obtained, and the amount of data necessary for processing and the associated processing time increases inadequately for this improvement.

In addition, the selection of these parameters should depend on the industry in which the scanner is to be used. For example, in the case of the clothing industry, the accuracy of the dimensions of 1 cm may be sufficient to determine the parameters of the figure in order to select the size of the clothing. On the other hand, if 3D models are to be used, for example, in the modeling industry, a high accuracy of reconstruction, especially of the face, is required.

When it comes to assessing the accuracy of the reconstruction, there is a correlation between the *RMSE* parameter from the “structure from motion” algorithm and the measurement uncertainty determined by the standard deviation of the mean σ from experimental comparison with the reference model. This correlation is not linear, but it allows to assess the accuracy of the reconstruction of the object in relation to other analyzed research variants at an early stage of data processing.

To sum up, for the best scanner configuration from the accuracy point of view, the *RMSE* parameter was equal to 0.77 [px], the mean error was μ= 0.65 [mm], while the standard deviation of the mean was σ= 0.65 [mm].

From the point of view of scanning efficiency, i.e., the relationship between the quality of the model and the processing time, the optimal variant seems to be c3, obtained for the same settings as the variant c1 = b1, but for a twice smaller number of photos. Therefore, it allows to shorten the time of data processing in order to obtain a 3D model.

## 5. Summary and Directions of Further Research

The selection of scanner parameters is a difficult task and requires multi-criteria optimization, taking into account e.g., the geometrical parameters of the scanner, the number and method of arrangement of the sensors around the object, the angular velocity of the rotating arm, the type and parameters of used sensors (in particular the resolution of the cameras), as well as the background and lighting conditions.

This work is mainly limited to the aspects related to the distribution of the cameras around the scanned object, and their impact on the accuracy of the reconstruction. Therefore, the dependencies describing the geometrical parameters of the scanner, including those related to the distribution of the cameras, were derived. A method of parametrization of the scanner’s geometrical configuration was proposed, which allowed for a significant reduction of the number of degrees of freedom in the selection of geometrical parameters. In addition, the derived dependencies are finally dependent on the height of the scanned object, thus they ultimately allow the scanner to be set up to the height of the scanned person.

The proposed method of selection of the geometrical parameters of the scanner includes the following steps:1.Select constant scanner parameters that enable the scanning of objects with the maximum height H and within the radius $r_{0}$ (see Figure 4 and Table 1);2.Refine the distance of the mast from the axis of rotation L and the angles of the deviation of the mast arms from the vertical $\gamma_{1}$ and $\gamma_{2}$, which guarantee similar distances of all cameras from the human figure $d_{i}$, guarantees sufficient overlap for photos and provides the smallest possible scanning space;3.For the known height of person H select the best camera distribution strategy, defined by the coefficients $dh_{i}$, and determine geometrical parameters of the scanner, including: $r_{i},\;z_{i}$, $\theta_{i},a_{i}$, $b_{j}$, $\alpha_{i}$, $\beta_{j}$;4.Select the minimum necessary number of camera poses around the scanned object n and vertically k, which for the used type of cameras provides the required accuracy of the object reconstruction for a given application, defined as the measurement uncertainty $\sigma_{M}$;5.Determine the remaining parameters describing the angle of view of cameras, that is $z_{vi}^{0}$, $z_{vi}^{min}$, $z_{vi}^{max}$ and the minimum distances from the assumed human figure $d_{i}$;6.Visualize the geometrical configuration of the scanner for the final parameters and the obtained 3D model.

As part of the preliminary research, numerical analyzes were carried out, which allowed for a further limitation of the choice of geometrical parameters of the scanner, taking into account the shape of the human figure. In particular, the configuration of the scanner arms was selected in such a way that the distances of the cameras from the human figure were approximately even.

In order to quantify the accuracy of the reconstruction, an accurate reference model was created with the use of a precision Hexagon Stereo Scan scanner. A specially prepared dummy was used in experimental research, which allowed the object to remain stable between tests. It also made it possible to quantify the scanning accuracy with the tested HUBO scanner for different strategies, in terms of the distribution of camera viewpoints along the height of the object, as well as a different number of cameras around the object and along the mast (vertically).

The accuracy of the object reconstruction was assessed using the *RMSE* parameter available in the AliceVision framework and the measurement uncertainty for the selected characteristic points of the model. This allowed for an objective selection of the preferred scanner configurations in terms of distribution and number of cameras. A correlation of the results in terms of the *RMSE* parameter and measurement uncertainty was also noticed. It follows that it is possible to perform a preliminary and quick assessment of the reconstruction accuracy already at the structure from motion stage, before determining the 3D model in the form of a dense cloud of points or a triangle mesh.

Data processing times were also recorded for each variant of the scanner configuration. This allowed for the assessment of the effectiveness of the individual scanner configurations, i.e., the accuracy of the reconstruction in relation to the processing time.

As a result of the experimental research, using the developed method, the influence of the selected parameters on the accuracy of the object reconstruction was analyzed. When analyzing the final results, it can be seen that good reconstruction accuracy was obtained for several scanner configurations. The optimal settings depend on the industry in which the scanner is to be used, as the accuracy of the reconstruction is not always of the same importance. In some industries, more attention is paid to processing time, so from this point of view, scanner configurations for fewer camera poses may be preferable.

As part of further work, it is necessary to continue searching for optimal settings for a wide range of heights of the objects. Apart from tests with the use of dummies, it is also necessary to perform tests in real conditions, i.e., with the participation of people. For people of lower height, decreasing the number of cameras along the height should be considered, so as not to reduce the accuracy of the reconstruction, but to reduce the dataset and shorten the processing time. As no reference model will be available for experimental studies with the participation of people, the *RMSE* parameter can be used to assess accuracy of reconstruction.

In terms of multi-criteria optimization, it is possible to investigate the influence of other factors on the accuracy of the reconstruction, such as the illumination of the object, or special treatment of the background, e.g., adding fiducial markers.

One can also consider the use of surface smoothing algorithms and artificial intelligence methods to eliminate the imperfections of the 3D model, especially its surface.

In turn, in order to shorten the data processing time, it is possible to take into account the use of incremental algorithms that could create a 3D model on the fly during scanning, using a few neighboring photos instead of the entire dataset.

## Figures and Tables

**Figure 1 sensors-22-09181-f001:**
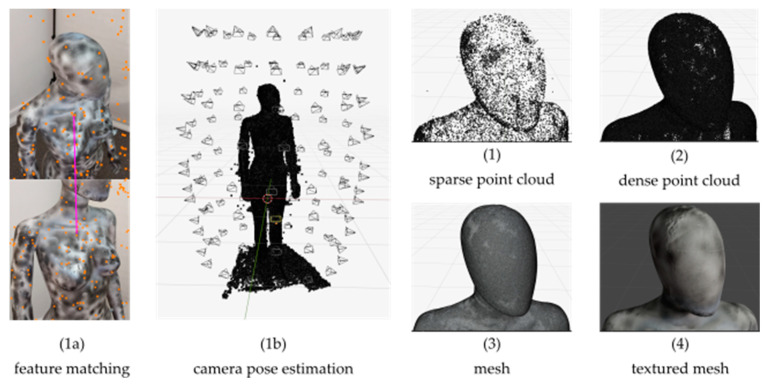
Illustration of stages of the reconstruction process using photogrammetry technique.

**Figure 2 sensors-22-09181-f002:**
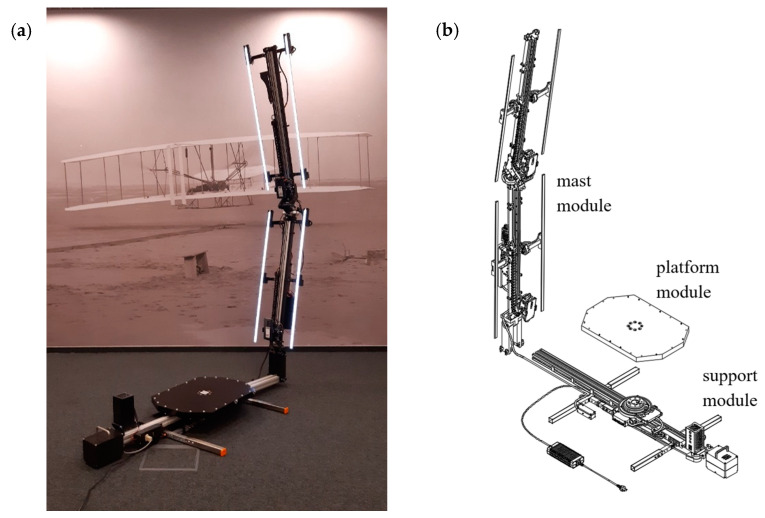
HUBO scanner: (**a**)—scanner design, (**b**)—illustration of the modular structure of the scanner.

**Figure 3 sensors-22-09181-f003:**
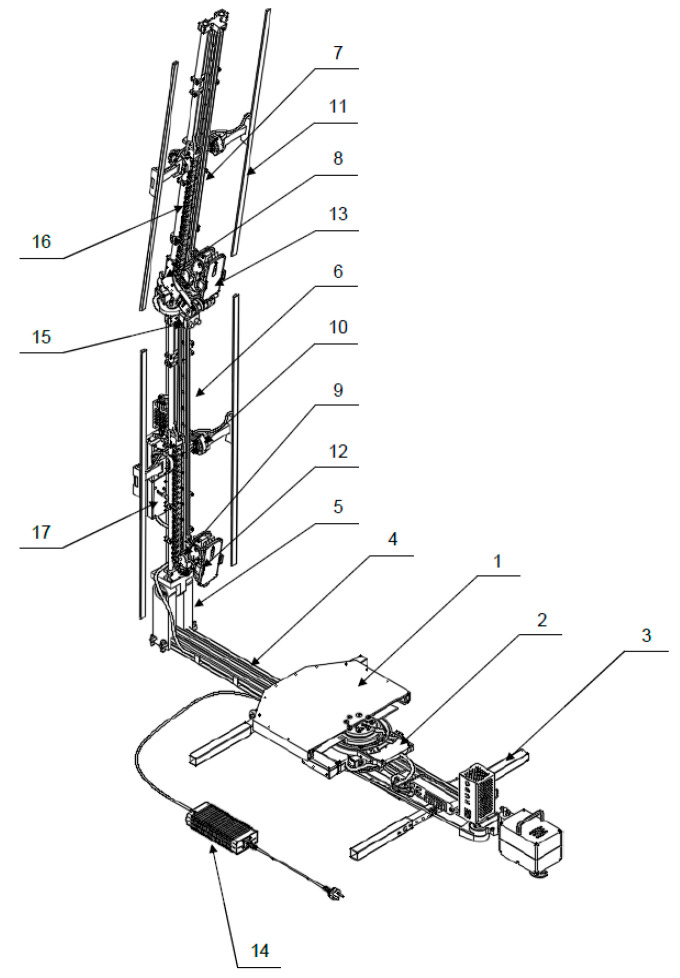
Illustration of the detailed design of the HUBO scanner.

**Figure 4 sensors-22-09181-f004:**
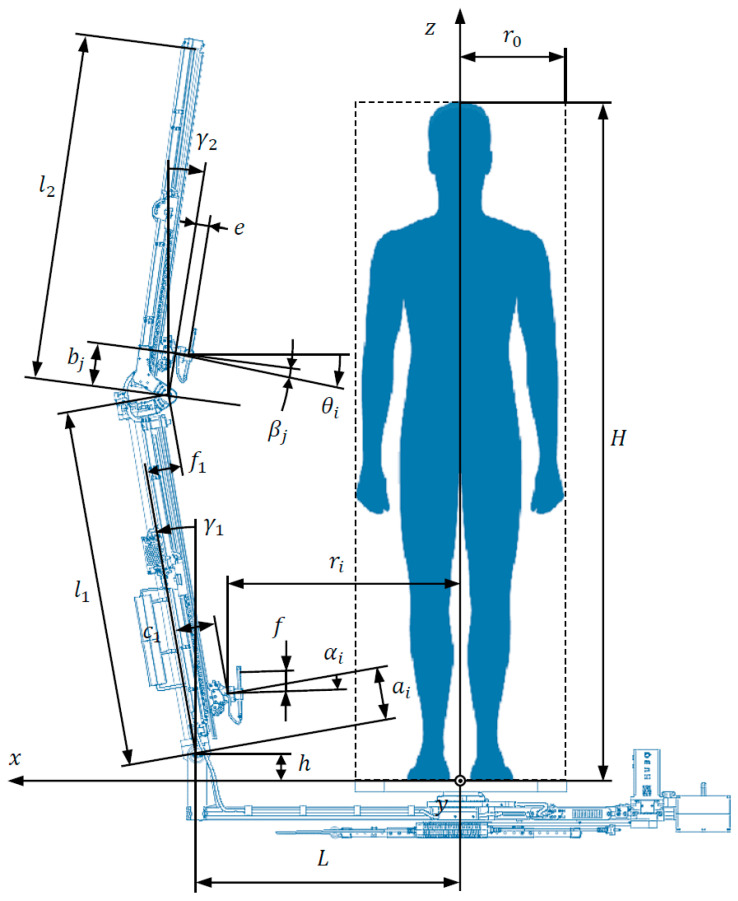
Geometrical parameters of the HUBO scanner and the scanned human.

**Figure 5 sensors-22-09181-f005:**
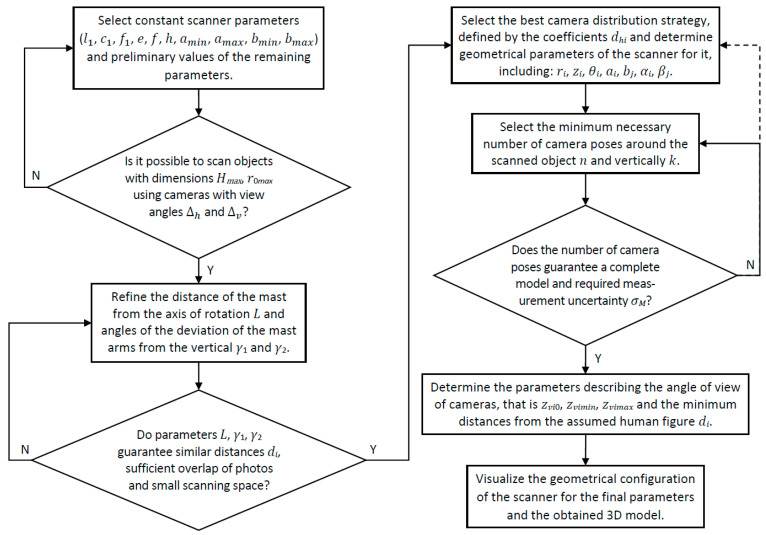
The method of selecting the geometrical parameters of the scanner.

**Figure 6 sensors-22-09181-f006:**
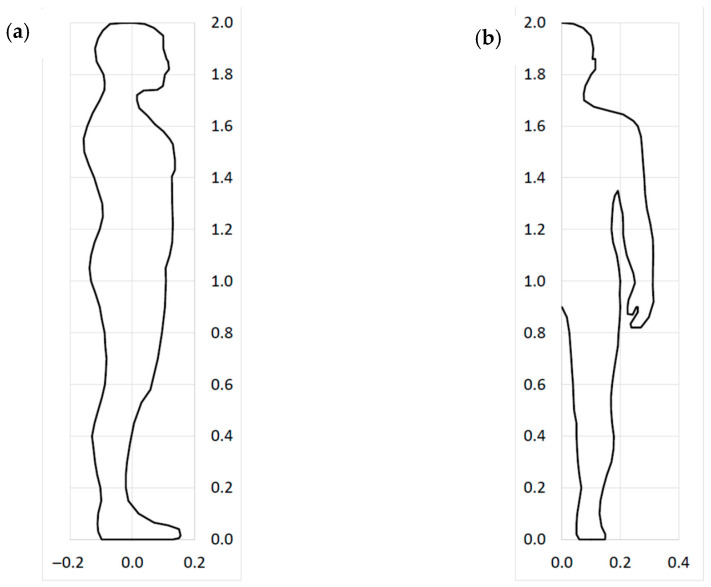
Human figure defined as a scalable set of points in the scanner coordinate system: (**a**) in the xz plane, (**b**) in the yz plane.

**Figure 7 sensors-22-09181-f007:**
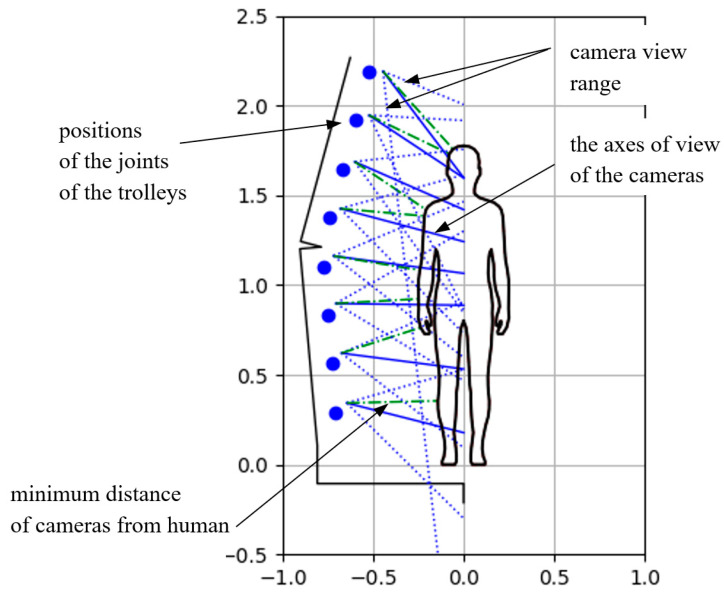
Illustration of the distribution and ranges of view of the HUBO scanner cameras in the yz plane.

**Figure 8 sensors-22-09181-f008:**
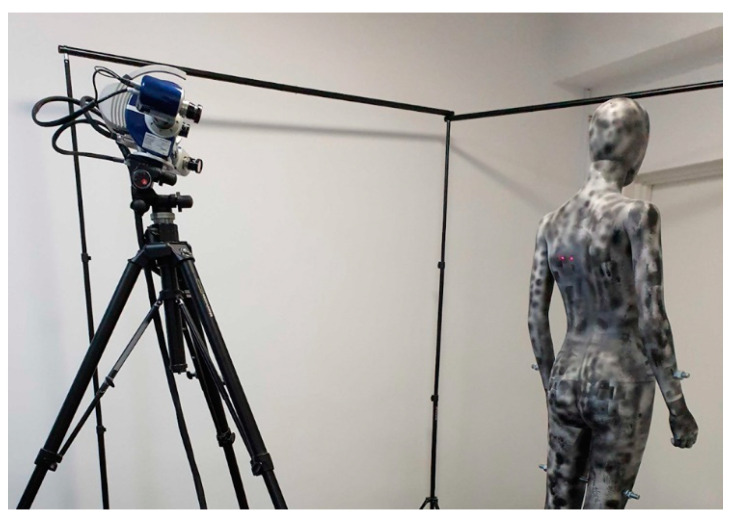
Determination of the reference model using the Hexagon Stereo Scan scanner (AICON).

**Figure 9 sensors-22-09181-f009:**
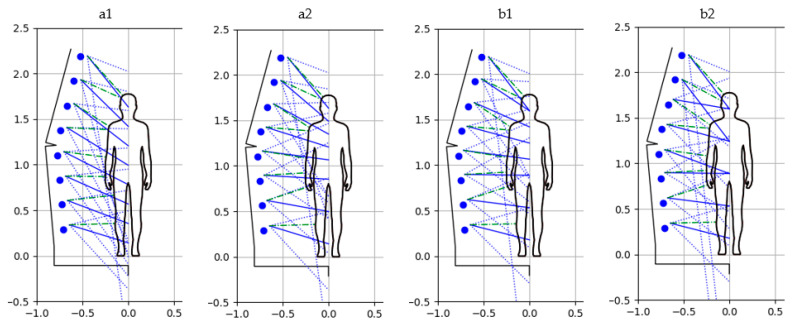
The illustration of camera configuration and visibility for particular variants.

**Figure 10 sensors-22-09181-f010:**
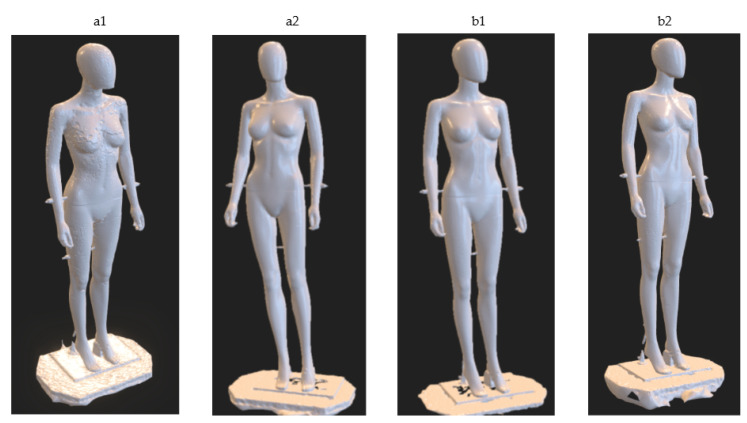
The results of the experimental tests of the HUBO scanner for particular variants—3D models in the form of a triangle meshes obtained for variants a1, a2, b1 and b2.

**Figure 11 sensors-22-09181-f011:**
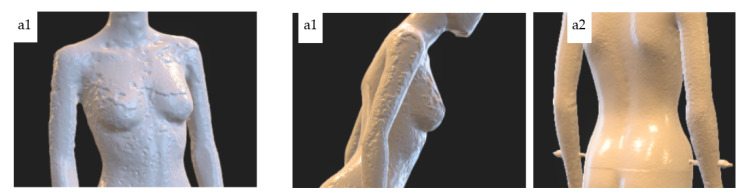
Selected characteristic fragments of 3D models for variants a1 and a2.

**Figure 12 sensors-22-09181-f012:**
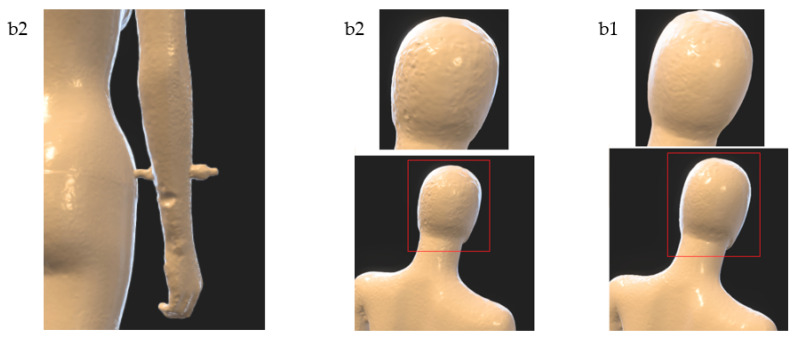
Selected characteristic fragments of 3D models for variants b1 and b2. Details in red rectangles are shown magnified.

**Figure 13 sensors-22-09181-f013:**
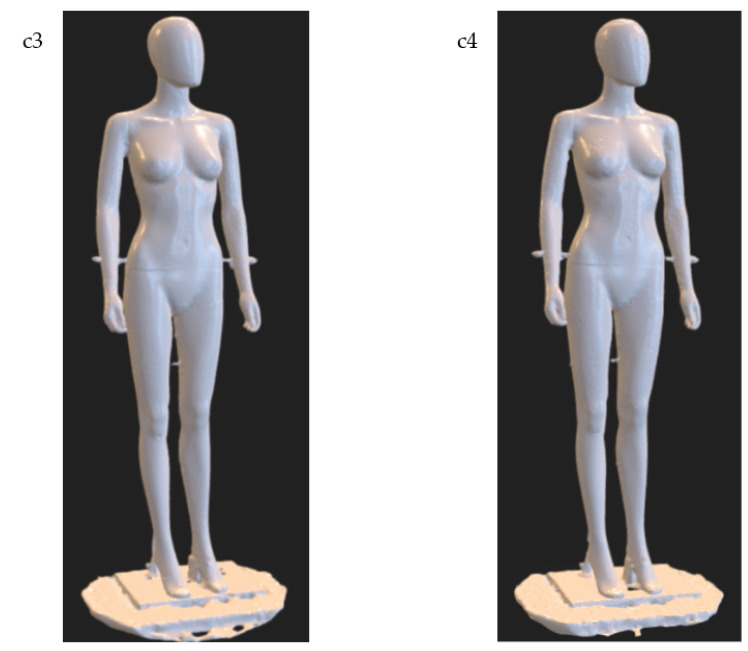
The 3D models in the form of a triangle mesh obtained for variants c3 and c4 of the experimental tests of the HUBO scanner, respectively.

**Figure 14 sensors-22-09181-f014:**
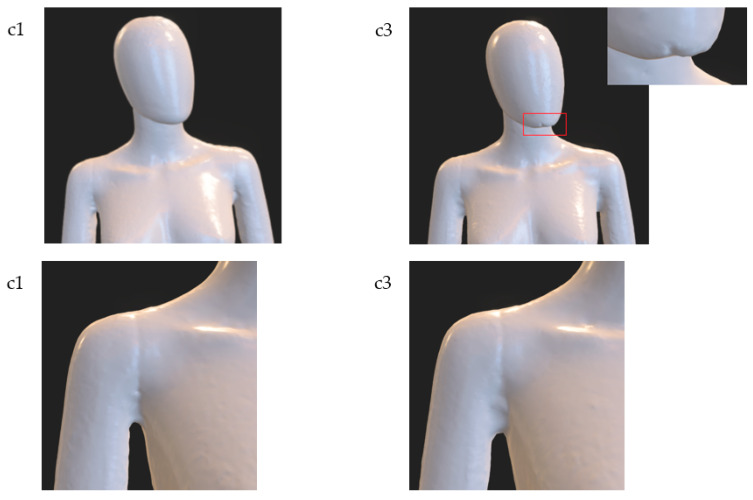
Selected characteristic fragments of 3D models for variants c1 and c3, respectively. Detail in red rectangle is shown magnified.

**Figure 15 sensors-22-09181-f015:**
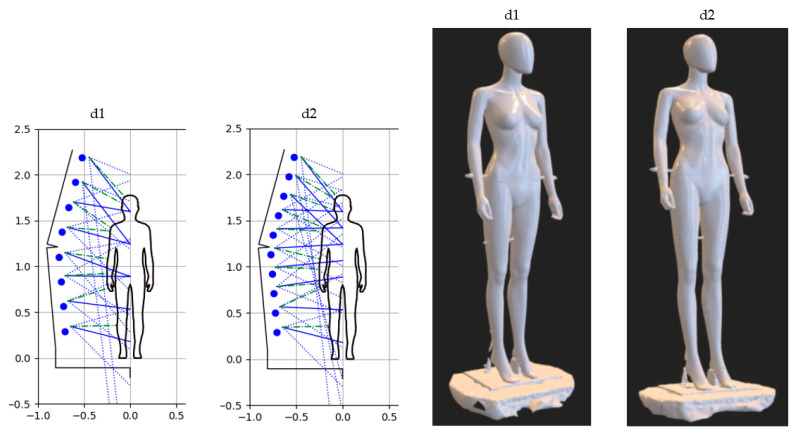
The illustration of camera configuration and visibility as well as the results of the experimental tests of the HUBO scanner for variants d1 and d2.

**Figure 16 sensors-22-09181-f016:**
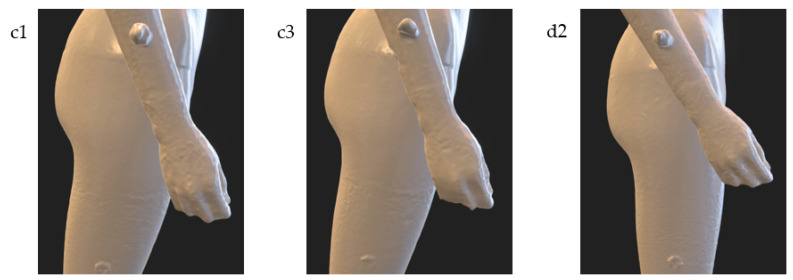
The illustration of reconstruction accuracy of hand for selected experimental tests of the HUBO scanner.

**Figure 17 sensors-22-09181-f017:**
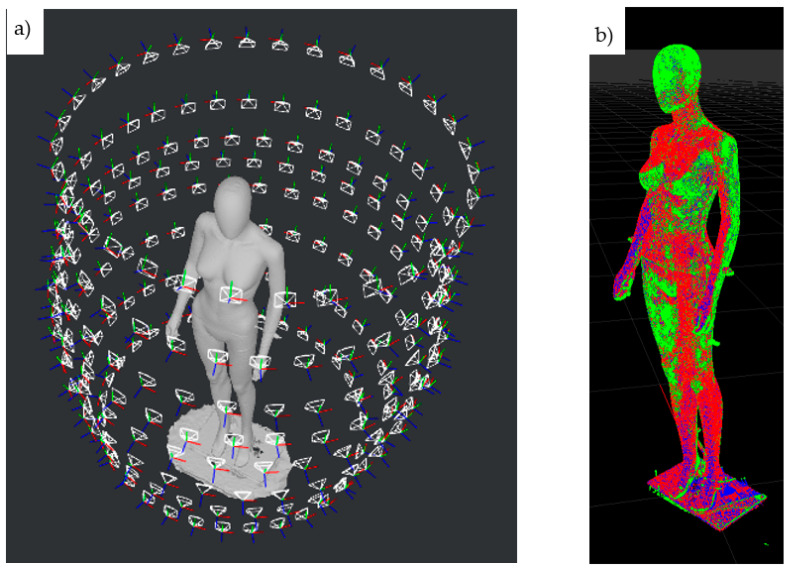
Sample results of dummy scanning with the use of HUBO scanner: (**a**) distribution of cameras around the scanned object obtained from the “structure from motion” algorithm, (**b**) the effect of comparing the scanning result with the reference 3D model.

**Table 1 sensors-22-09181-t001:** Constant geometrical parameters of the HUBO scanner adopted for the research.

$l_{1}$ [m]	$c_{1}$ [m]	$f_{1}$ [m]	e [m]	f [m]	h [m]	$a_{min}$ [m]	$b_{max}$ [m]	$b_{min}$ [m]	$b_{max}$ [m]	$\Delta_{h}$ [°]	$\Delta_{v}$ [°]
1.100	0.119	0.119	0.045	0.065	0.106	0.18	0.95	0.14	1.01	36.5	61

**Table 2 sensors-22-09181-t002:** Geometrical parameters characterizing the arrangement of the HUBO scanner cameras for an exemplary numerical analysis.

Quantity and Its Unit	Camera Number							
	1	2	3	4	5	6	7	8
$r_{i}$ [m]	0.706	0.729	0.753	0.773	0.743	0.670	0.597	0.525
$z_{i}$ [m]	0.291	0.562	0.833	1.104	1.375	1.646	1.917	2.188
$d_{i}$ [m]	0.504	0.461	0.459	0.471	0.468	0.495	0.491	0.582
$\theta_{i}$ [°]	−14.3	−7.5	−0.8	−20.3	−15.1	−9.7	−52.6	−53.1
$a_{i},\;b_{j}$ [m]	0.175	0.447	0.719	0.950	0.169	0.449	0.730	1.01
$\alpha_{i}$, $\beta_{j}$ [º]	−19.3	−12.4	−5.8	−25.3	−0.1	5.3	−37.6	−38.0

**Table 3 sensors-22-09181-t003:** The quantitative comparison of the research results.

Variant	*RMSE* [px]	μ [mm]	σ [mm]	T [mm]	*EF* [-]
a1	1.03	1.24	1.03	123	127.0
a2	0.85	0.86	0.71	103	73.1
b1 = c1	0.77	0.65	0.65	101	65.3
b2 = c2 = d1	0.85	1.02	0.81	105	84.9
c3	0.83	0.69	0.66	63	41.6
c4	0.92	1.02	0.84	64	54.0
d2	0.96	0.81	0.76	156	118.1

## Data Availability

Publicly available datasets were analyzed in this study. This data can be found here: https://bit.ly/3tYrASJ.

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
