# Peer review of "Analysis of the Influence of the Geometrical Parameters of the Body Scanner on the Accuracy of Reconstruction of the Human Figure Using the Photogrammetry Technique"

_sensors, 2022, doi:10.3390/s22239181_

Round 1

Reviewer 1 Report

This paper proposes a method for selection of geometrical parameter of a 3D body scanner in order to obtain the most efficient and acceptable accuracy model.

Concerning the paper organisation, its ok, however in the intro regarding the description of the three stages, I suggest to put a set of figures complemented with parts of the text to better illustrate the steps and what are they outputs, as example the 3d mesh formation. To clarify reader about those.

Regarding the problema of minimum distance of he cameras and the scanned figure, how this can be overcome with cameras with larger resolution. The higher resolution would enable to relax a bit this minimum distance criteria. This distance reduction also impact directly the degree of overlap of cameras, requiring the need of more cameras to cover the volume namely deeper cut regions. Of course in regions with curvatures this has a strong impact. Have you accesses this problematic or reached an optimal set configuration in a relation. Of curvatures vs camera density, bellow parts often require less camera coverage since they are mostly straight as example.

Also some more examples on the outputs of the several variant created artefacts would better illustrate the mais problematic and were are the most sensitivity parts of the body that require more camara density.

I suggest to rearrange the bullet points/item to be better readable, they are a bit messy. For example this regions:

B. selection of the optimal arrangement of cameras on the scanner arms, including:

397

a. number of cameras, taking into account:

1. number of positions around the scanned object ?,

2. number of positions along the mast arms ?,

b. positions of the cameras on the mast arms ? ?and ? ?,

c. pitch angles of the cameras relative to the mast arms ??

and

? ?.

Reviewer 2 Report

see attached file for comments
